# Cardiac and Mental Benefits of Mediterranean-DASH Intervention for Neurodegenerative Delay (MIND) Diet plus Forest Bathing (FB) versus MIND Diet among Older Chinese Adults: A Randomized Controlled Pilot Study

**DOI:** 10.3390/ijerph192214665

**Published:** 2022-11-08

**Authors:** Ka-Yin Yau, Pui-Sze Law, Chung-Ngok Wong

**Affiliations:** 1School of Nursing, Tung Wah College, Ho Man Tin, Hong Kong; 2School of Nursing, and Health Studies, Hong Kong Metropolitan University, Ho Man Tin, Hong Kong; 3Research Office, Tung Wah College, Ho Man Tin, Hong Kong

**Keywords:** MIND diet, forest bathing, hypertension, cardiovascular risk factor, mood state, anxiety

## Abstract

(1) Background: The Mediterranean-DASH intervention for neurodegenerative delay (MIND) diet and forest bathing (FB) are first-line therapies for controlling hypertension. This study aimed to investigate the combined effects of a MIND diet and FB and MIND diet alone among older Chinese patients with hypertension. (2) Methods: Seventy-two participants aged >50 with stages 1 or 2 hypertension were randomly assigned to the MIND group (*n* = 23), MIND-FB group (*n* = 25), or control group (*n* = 24). Systolic blood pressure (SBP) (primary outcome), point-of-care tests for blood lipid panel and glucose (Glu), anxiety levels, mood states, body mass index (BMI), waist-to-hip ratio (WHR), and body fat percentage (BFP) were measured. (3) Results: After a four-week intervention, the change in SBP revealed no significant differences between the intervention groups, and SBP tended to decrease in the MIND and MIND-FB groups. Total cholesterol, and low-density lipoprotein (LDL-C) were significantly decreased (*p* < 0.01), Triglycerides (TG) and Glu levels were significantly lower (*p* < 0.05) in the MIND-FB and MIND groups, and the mean differences for anxiety level and negative mood states were significantly lower (*p* < 0.00) in MIND-FB group. (4) Conclusions: The results provide preliminary evidence that the MIND diet and FB are good for promoting cardiac and mental health well-being in the Chinese population.

## 1. Introduction

Hypertension is a well-known risk factor for cardiovascular disease (CVDs) [1]. High blood pressure (BP) has caused 235 million disability-adjusted life-years and 10.8 million deaths in 2019 globally [2]. The American Heart Association defined stage 1 hypertension as systolic BP (SBP) 130–139 mmHg or diastolic BP (DBP) 80–89 mmHg, and stage 2 as SBP ≥ 140 mmHg or DBP ≥ 90 mmHg [3]. Adults with hypertension were reported to have an increased risk of CVDs with age and whether they have other modifiable risk factors, including unhealthy diet, elevated lipid and glucose, overweight, physical inactivity, and under stress [1,4]. Regular antihypertensive drug therapy was recommended for individuals with hypertension to control the potentially impaired BP. However, they were reported to not comply with their prescribed antihypertensive regime or to discontinue medication for the adverse effects and the high cost of medicine [5,6]. It increased their risk of cardiovascular morbidity and mortality [6,7]. Lifestyle modification, the first line of hypertension therapy, is effective in controlling BP, preventing CVDs incidence and mortality with weight loss, adopting Dietary Approaches to Stop Hypertension diet (DASH) or Mediterranean diet (MedDi), regular physical activity, a reduction in alcohol consumption and regular stress-relief practice [3,8,9,10,11].

Evidence has shown that adopting DASH and MedDi diets contributes to cardiovascular health [3,8,9,10,11]. Promoting a culturally tailored healthy diet pattern is necessary due to cultural preferences, different geographic settings, and health conditions. The Mediterranean-DASH Intervention for Neurodegenerative Delay (MIND) diet [12] is a hybrid of the MedDi and DASH diet, MIND diet emphasizes the intake of more fruits and green leafy vegetables, berries, and olive oil which contribute to controlling body weight, suppressing oxidative stress, and preventing metabolic syndrome [13,14,15]. Recently, a study found that the MIND diet score was associated with lower odds of reduced HDL cholesterol and general obesity [16]. Apart from cardioprotective effects, a recent study showed that adopting a MIND diet effectively reduced depression among older adults [17].

Shinrin-Yoku refers to forest bathing (FB), which is a Japanese practice of relaxing using the five senses by immersing oneself in nature. A review of 14 experimental studies was performed, of which 11 measured BP as the primary outcome. Six of the studies reported that FB significantly decreased in SBP by 5.4–24.6% and DBP by 7–29.5% after 2 h single forest visit or a day FB program [18]. Another meta-analysis study revealed that visiting a forest environment significantly lowered SBP (−3.15 mmHg) and DBP (−1.75 mmHg), respectively, compared to a non-forest environment [19]. Studies have indicated that FB is effective in reducing BP and anxiety levels and enhancing the positive mood of participants, and FB could be suggested to manage and control hypertension [18,19,20].

Adopting a MIND diet can have beneficial effects on cardiovascular health for people living in a metropolis such as Hong Kong. The brain-healthy food components of the MIND diet could protect arteries against obesity and depression through their antioxidant and anti-inflammatory properties [17,21]. The MIND diet’s cardioprotective and mental protective effects have not been examined or compared in Asian populations. Regarding the differences in the food supply, food consumption pattern, and lifestyle across Eastern and Western countries, studies on MIND are required in Asian and Chinese populations. FB is suitable for older people to relax because of its low-intensity level of exercise. The possible therapeutic effects of the MIND diet and FB are associated with lowering BP and reducing stress among adults with hypertension. Studies on the synergistic effect of MIND and FB are required to control hypertension. Few studies have examined the synergistic effect of dietary control and stress management on hypertension control [22,23,24]. To the best of our knowledge, no study has been conducted in the local community. Therefore, we hypothesized that the MIND diet combined with FB could have more cardiac and mental benefits in the older Chinese hypertensive population than the MIND diet alone.

This study aimed to examine and compare the effects of a four-week period of MIND diet plus FB, MIND diet only, and control intervention on systolic BP, lipid profile and glucose, waist-to-hip ratio, body fat percentage, body mass index, state and trait anxiety levels, and mood states of individuals with hypertension. More specifically, we wish to detect the feasibility of such a project concerning the appropriateness of the intervention plan and the efficacy of the outcome measures that should be addressed in the design of future trials on this topic.

## 2. Materials and Methods

### 2.1. Study Design

A three-group RCT design with equal allocation across groups (1:1:1) was used. The three arms were MIND plus FB and MIND alone as the intervention groups and usual care as the control group. The participants generated a code by lucky draw, randomized following block randomization procedures with a block size of 6 or 12. This was a single-blind study in which assessors were blinded to treatment assignment.

### 2.2. Participants and Selection Criteria

The participants were older adults with hypertension recruited from two local community centers. A designed invitation poster was posted on the network of the two community centers between November and December 2021. Potential participants completed an online demographic questionnaire and were screened for BP and health history at baseline assessment to determine eligibility. Potential participants who met the inclusion criteria were asked to sign a consent form before participating in this pilot study. Inclusion criteria were (1) age > 50 years; (2) Chinese ethnicity; (3) ability to speak and understand Chinese; (4) fulfilment of the American Heart Association (AHA) criteria for stage 1 (SBP 130–139 mmHg or DBP 80–89 mmHg) and stage 2 hypertension (SBP ≥ 140 mmHg or DBP ≥ 90 mmHg); and (5) physical fitness to participate in mild-intensity exercises. Exclusion criteria were (1) poorly controlled respiratory and kidney disease; (2) allergy to nuts, berries, olive oil, or fish; (3) known mental disorder; (4) pregnancy; (5) those who were currently participating in any dietary or relaxation program or those who just completed a dietary or relaxation program within 3 months; (6) those who experienced chronic muscle weakness and pain; (7) limited walking; and (8) extremely high SBP > 159 mmHg and/or DBP > 100 mmHg.

### 2.3. Sample Size

No previous RCT has been conducted on the MIND diet. Based on the results of systolic BP as a primary outcome variable of a DASH trial [25], it was assumed that there was a large and moderate improvement (in terms of reduced SBP) in the MIND plus FB and MIND groups, respectively, and no improvement was observed in the control group (Cohen d effect size of 0.7). To assess the 5% level of significance (two-sided test) of the effects among three interventions (MIND plus FB, MIND only, and control) on the primary outcome (i.e., reduction in SBP) in participants, with a power of 80%, dropout rate 10%, a total of 78 participants will be recruited for this pilot study (26 per arm). The sample size was estimated based on the F-test using G*Power 3.1.9.4.

### 2.4. Intervention and Control

#### 2.4.1. MIND Diet Intervention

Participants in the MIND diet group were instructed to consume the MIND diet for 4 weeks. They received a 4-week nutrition program consisting of four hybrid mode sessions (one session per week, 60 min per session, face-to-face or online mode) to modify their diet. Each intervention session was conducted by a nutritionist and research assistant. The nutritionist-to-participant ratio in each session was 1:15. The topics covered included an overview of the MIND diet, a discussion for establishing rapport with participants, and instructions regarding BP measurement, dietary adoption, and adherence to nutritional recommendations. The strategies for promoting readiness for changes in the nutrition session included exploring the participants’ capability, opportunity, and motivation to adopt the MIND diet.

#### 2.4.2. MIND-plus-FB Intervention

The participants in the MIND-plus-FB group received the same intervention as the MIND group and 2 h FB sessions on 4-consecutive weekends in the Quarry Bay tree walk. The Quarry Bay tree walk, located in the eastern corner of Hong Kong Island, was an important military base during World War II. The area overlaps with part of Wilson Trail Section 2 and the Eastern Nature Trail, intersecting Mount Parker Road and Kornhill. The trail covers 270 hectares of plantations reaching heights of 2–10 m, including Taiwan acacia, Polyspora axillaris, and Ormosia semicastrata; flowering plants include Gordonia axillaris, and wild plants, such as Melatoma candidum and Camellia hongkongensis [26]. The average air temperature and relative humidity were 18 °C and 72.4%, respectively, on each day the FB session took place. The total distance covered was <0.5 km with a gentle slope of 5% and the altitude of 100 m. The average walking speed was around 0.25 km/h.

The participants gathered at the country park entrance on the day of FB. The FB coach then began the FB intervention using a set of experiential thresholds via sensory activity to guide participants in immersing themselves in nature deeply, followed by an identical invitation: (i) sharing experience for the forest, (ii) pleasure of moment using five senses, (iii) slowing down both physically and mentally, and (iv) building up connections to the forest. Participants were encouraged to rest and share the time between each change in activity (Table 1). The tea ceremony was incorporated at the end of each session as the second threshold. Each session was guided by a certified FB coach with a wilderness first aid certificate from the Association of Nature & Forest Therapy (ANFT) and a research assistant (holding a first aid license) to ensure safety and maintain consistency during FB. The coach-to-participant ratio was approximately 1:7 for each group for a total of two groups for each session (Figure 1).

#### 2.4.3. Control Group

Participants in the control group were instructed to perform usual lifestyle activities during the intervention periods.

### 2.5. Primary and Secondary Outcomes

The primary outcome was systolic BP. Systolic BP was determined using a validated digital automatic sphygmomanometer (Omron M6 comfort) [27]. BP was measured twice in a sitting position after resting for at least for 10–15 min and based on the mean of two measurements of a participant.

The secondary outcomes were the POCT of the lipid panel [total cholesterol, HDL-C, Triglycerides (TG), and LDL-C] and Glucose (Glu), body mass index (BMI), waist-to-hip ratio (WHR), body fat percentages (BFP), anxiety level, and mood states. The POCT of the lipid panel was measured using the Cholesterol Reference Method Laboratory Network (CRMLN) certified CardioChek PA [28]. The participants were instructed to fast for no less than 6 h. BMI and body fat percentage were measured using bioelectrical impedance analysis (BIA). Waist circumference was measured between the lower rib and iliac crest using a tape meter. The waist-to-hip ratio was calculated as waist circumference (cm) divided by hip circumference (cm). Anxiety level and mood states were measured using self-reported Chinese questionaries of State and Trait Anxiety Inventory (STAI) and The Profile of Mood States Short Form (POMS-SF) (alpha coefficient ranging from 0.9 and 0.81 and 0.98 to 0.99, respectively) [25,29,30].

The outcome measurements were recorded in the evening (4 pm to 8 pm) before the intervention (at the baseline assessment) (T0) and after the 4-week intervention (4th week) (T1) by trained research assistants to determine its short-term effects. Sociodemographic data (age, marital status, sex, educational attainment, and living arrangement), physical activity, alcohol consumption, BMI, medical diagnosis, current drug use, and dietary intake were obtained at the baseline assessment. Dietary intake was assessed by using a validated 163-item semi-quantitative Harvard Food Frequency Questionnaire (FFQ) [31,32]. The MIND dietary pattern was assessed by calculating summation scores using predefined criteria [12]. The total MIND diet score was computed by summing all food groups; 15 had the highest scores, representing the highest adherence to the MIND diet.

### 2.6. Statistical Methods

All statistical analyses were performed using SPSS version 26 software (SPSS Inc., Chicago, IL, USA). Intention-to-treat (ITT) analysis was used to replace the missing values with the last-observation-carried-forward principle. A per-protocol analysis was performed for sensitivity analysis. The results are shown in Appendix A. Pearson’s X2 tests and analyses of variance (ANOVA) were used to compare categorical and continuous variables at baseline and 4 weeks. The paired *t*-test, one-way ANOVA, Tukey’s honest significant difference (HSD), and ANCOVA model with post hoc test were used to analyse the mean difference between and among groups, respectively. The effect size was calculated using the r2 coefficient of determination. Statistical significance was set at *p*-values of <0.05.

## 3. Results

### 3.1. Study Population

This study was conducted from November 2021 to April 2022 in Hong Kong. Seventy-two participants were recruited. Of these, 70 participants (97%) completed the study. Figure 2 shows the flow of participants throughout the trial.

### 3.2. Baseline Characteristics

Sociodemographic and lifestyle characteristics of participants are shown in Table 2. The mean age of participants was 66.9 +/− 9.7 years, and 76.4% were female. Half (47.2%) of the participants had completed secondary education, 45.8% performed exercise 1–2 times a week, and 31.9% performed exercise >40 min per week. Most participants reported no smoking (97%) or drinking habits (90%). Furthermore, 56.9% of participants reported hypertension with medication. A higher proportion of participants in the Control Group were elderly, aged 65 or above (79.2%), compared to 60.9% in the MIND Group and 44% in the MIND-FB Group (*p* = 0.041). No significant difference was found among the three groups in the presence of hypertension and antihypertensive medications intake.

The baseline measurements of the participants are shown in Table 3. The participants’ mean waist-to-hip ratio and diastolic BP differed among the three groups.

### 3.3. The Intervention Effect on BP

Table 4 shows the change in interventional outcomes from baseline to week 4 of the intervention and effect size. Although not statistically significant, the SBP in MIND and MIND-FB groups decreased slightly after the four-week intervention (with a mean decrease of 1.93 mmHg and 3.18 mmHg respectively), compared to a slight increase of SBP in the Control group (with a mean increase of 0.66 mmHg). A similar result was found when we adjusted the model for age, waist-to-hip ratio, and SBP at baseline.

After the four-week intervention, a decrease in DBP was found in the MIND-FB group (with a mean reduction of 2.57 mmHg, *p* = 0.036), compared to a slight decrease of DBP in the MIND group (mean reduction = 1.26, *p* = 0.429) and the Control group (mean reduction = 0.27, *p* = 0.798). The mean difference in SBP at week 4 of the intervention was not statistically significant among the three groups after adjusting for age, waist-to-hip ratio, and SBP at baseline (Figure 3).

### 3.4. Intervention Effect on Other Variables

The changes in adiposity, cardiovascular risk factors, and psychological indicators in each group and the mean differences between the three groups after the four-week intervention are shown in Table 4. After the four-week intervention, the mean waist-to-hip ratios for the MIND group, MIND-FB group, and control group were 0.885 (95% CI: 0.86–0.91), 0.89 (95% CI: 0.87–0.91), and 0.92 (95% CI: 0.90–0.94), respectively. In a post-hoc comparison, the mean waist-to-hip ratio for the MIND group was 0.003 less than that of the control group (*p* = 0.79). Similarly, the mean waist-to-hip ratio for the MIND-FB group was 0.012 less than that of the control group (*p* = 0.28). The mean BMI demonstrated a significant decrease of 0.24 kg/m^2^ in the MIND-FB group compared to that of the control group. Although all cardiovascular risk factors, trait anxiety levels demonstrated no significant difference between groups, total cholesterol, and lower-density lipoprotein cholesterol (LDL-C) were significantly decreased by 0.61 and 0.34 mmol/L (*p* < 0.01) in the MIND group; 0.86 and 0.59 mmol/L (*p* < 0.01) in the MIND-FB group, and 0.58 and 0.28 mmol/L (*p* < 0.02) in the control group, respectively. Triglycerides (TG) and glucose (Glu) levels were significantly lower at 0.28 mmol/L and 0.68 mmol/L (*p* < 0.05) in the MIND-FB and MIND groups, respectively. The mean difference for state and trait anxiety level, total mood states, tension–anxiety, fatigue, anger, and confusion were significantly lower by 5.28, 4.28, 6.5, 2.2, 1.4, 1.24, and 1.28, respectively (*p* < 0.00) in the MIND-FB group. The mean score of anger–hostility in the MIND_FB group demonstrated a significant 2.32 lower than that of the control group. Also, the mean score of stat anxiety levels in the MIND_FB and MIND groups was significantly lower than that of the control group, by 7.47 and 4 points, respectively. After the four weeks of intervention, the mean score of MIND diet consumption was significantly increased by 3.8 and 3.48 in the MIND and MIND-FB groups, respectively.

## 4. Discussion

The present study demonstrated that (1) adopting the MIND diet for four weeks might have cardioprotective and mentally protective effects among older Chinese patients with hypertension, (2) the MIND diet can be easily adopted by older Chinese individuals, and (3) FB could enhance MIND diet cardiac and mental benefits in older Chinese populations with hypertension.

Diet modification alone is highly recommended for individuals with elevated BP and stage I hypertension who do not qualify for initial antihypertensive drug therapy. Sodium restriction (<1500 mg/day); increased dietary potassium intake (3500–5000 mg/day); and a healthy diet such as DASH, which is rich in fruits, vegetables, whole grains, low-fat dairy products, lower saturated and total fat, and the MedDi diet, which refers to a high intake of olive oil, rich in green leafy vegetables and fruits, moderate fish and other meat, cereals, nuts, legumes/pluses, dairy products, low intake of eggs, red wine, and sweets, have been widely adopted for the prevention and management of hypertension [11,33,34]. Reduced dietary sodium intake, enhanced dietary potassium intake, and adoption of a DASH diet have been reported to reduce SBP by 5–6 mmHg, 4–5 mmHg, and 11 mmHg for individuals with hypertension [33]. Adopting a MedDi diet for one year could significantly reduce SBP by 1.5 mmHg in normotensive and mildly hypertensive individuals [34]. Recently, growing evidence has shown that the MIND diet, a combination of MedDi and DASH diets, contributed to a lower risk of developing Alzheimer’s disease and preventing heart disease [12,13,14,15,35,36,37]. The MIND diet emphasizes 10 brain-healthy foods (green leafy vegetables, other vegetables, berries, nuts, beans, whole grains, seafood, poultry, olive oil, and wine) and five unhealthy food groups (red meat, butter and stick margarine, cheese, pastries and sweets, and fried/fast food). No study has evaluated the effect of MIND diet alone on lowering high BP. In the present study, despite adopting the MIND diet alone for four weeks with four nutritional education sessions showing no SBP-lowering effect on older Chinese hypertension, adopting it demonstrated a tendency towards decreased SBP and DBP. The present study’s results revealed the MIND diet’s potential therapeutic effect in preventing high BP. Further studies are necessary to evaluate the short-term and long-term effects of the MIND diet alone on the management of hypertension among older Chinese patients with hypertension.

A review of 20 studies found that adopting DASH diet for 2–24 weeks could significantly reduce total cholesterol and LDL by 0.20 mmol/L (95% CI—0.31, −0.10; *p* < 0·001) and 0.10 mmol/L (95% CI—0.20, −0.01; *p* = 0.03), respectively, compared to control group [38]. However, another systematic review of 20 studies reported that the DASH diet did not have a beneficial effect on fasting blood glucose [39]. Two recent systematic reviews and meta-analyses have shown that MedDi is positively associated in preventing cardiovascular function [10,11]. The MIND diet has positively affected cognitive function since 2015 [12,13,14,15,36]. Few researchers have studied the beneficial effects of a MIND diet on cardiovascular risk factors [40]. The MIND diet emphasized the intake of more fruits and green leafy vegetables, berries, and olive oil but limited the intake of saturated fats, red meats, and added sugars. The recommended foods are a thorough source of fiber, antioxidants, phenolic compounds, and dietary fats, which contribute to weight control, suppress oxidative stress, and reduce the risk of developing metabolic syndrome. Berries and green leafy vegetables, in flavonoids, folate, vitamin E, and carotenoids, have been reported to have an antioxidant and anti-inflammatory effect in regard to inhibition of the formation of Aβ [40]. These mechanisms directly help with good brain health and indirectly impact the cardioprotective effects of related food components. The present study identified that adopting the MIND diet for four weeks could lower the total cholesterol, LDL-C, and blood glucose levels of Chinese older adults with hypertension. The diet pattern and lipid levels have changed significantly during the four-week period in the control group. The majority of the control group participants have borderline high total cholesterol and other lipids. The baseline screening may help to increase health awareness of the participants and in turn modify their diet patterns. The explanation for the screening results may also convey messages about their health to the control group. However, adopting the MIND diet did not show a potential effect on reducing anxiety levels or improving mood states in this pilot study. The current evidence of the MIND diet’s cardioprotective and mental protective effects is insufficient for the prevention and management of hypertension. Further research is necessary to explore the long-term and sustained effects of the MIND diet on cardiovascular and mental health in individuals with hypertension.

The MIND diet emphasizes the limited consumption of fast foods, fried foods, butter, margarine, pastries, and sweets [12]. These foods are commonly consumed by people living in urban areas. The MIND diet is culturally and contextually feasible for living in a metropolis such as Hong Kong. The MIND diet encouraged followers to increase their consumption of plant-based foods, fish and poultry, nuts, and berries, easily found in local markets and which people are likely to purchase. Moreover, the MIND diet focuses on daily and weekly recommendations for specific food and food groups. Qualified nutritionists guided the followers to determine how often the recommended foods in the diet should be consumed. For example, followers are advised to eat three or more servings of vegetables and minimally processed whole grains per day or more servings of berries and nuts per week. No guidelines are to be established following the MIND diet. Followers can build on their own daily or weekly eating patterns without tracking their daily calorie intake. This study’s higher MIND diet score reflected that older Chinese adult adopted the MIND diet eating pattern. A disadvantage of the MIND diet is that the recommended foods may not be appropriate for everyone to eat because of allergies and food preferences.

FB includes a low-intensity level of exercise for older people for relaxation. FB has been reported to have restorative effects on individuals connected with natural beings using the five senses [18,19]. FB has a beneficial effect on activating the parasympathetic nervous system through exposure to the natural environment and the subsequent stimuli of the five senses. The parasympathetic nervous activity is increased in turn to reduce the activity of the sympathetic nervous system, cortisol, and salivary amylase levels, causing a stage of relaxation [20]. FB helps reduce anxiety levels and negative moods among hypertensive older [18,19]. In addition, the MIND diet emphasizes the limited intake of fast foods and processed foods that are associated with the inflammatory response and the activation of cell-mediated immunity. Inflammation may increase the risk for neuroprogression in patients with depression. Adopting the MIND diet has been reported to have protective effects on reducing the risk of depression among older adults [35]. The results of the present study showed that consumption of the MIND diet for four weeks had no significant effect on reducing anxiety and negative mood. Although four weeks of MIND-FB intervention demonstrated no significant lowering effect on SBP, it could significantly lower cardiovascular risk factors. Besides its cardioprotective effect, MIND-FB improved mood states and reduced anxiety levels in older Chinese individuals. The present study showed that the MIND diet combined with FB could promote cardiac and mental health among the older Chinese hypertensive population more than the MIND diet alone.

### Limitation and Implication

The generalizability of this study was limited. The participants were recruited from two community health centers. They cannot represent all hypertensive populations. Moreover, due to the nature of the intervention, participants were randomly assigned to groups, but they were not blinded to group assignment, which may have caused bias. Larger and more representative samples are recommended in future studies to minimize the risk of bias.

The major findings of this study were that the MIND diet alone and MIND-FB were associated with a lower blood cholesterol and triglyceride after the completion of 4 weeks of intervention. Practicing FB can enhance the cardioprotective effect, reducing anxiety levels and negative moods among older Chinese individuals with hypertension. The MIND diet combination with FB, provides an alternative approach for Chinese older individuals with hypertension on its management. Since the MIND diet has not been widely adopted in Asia, relevant studies have only focused on its neuroprotective effect. It is recommended that researchers conduct more RCT trials to prove the therapeutic effect on cardiac and mental health. Since the lowering effect of the MIND diet alone and MIND-FB on total cholesterol level and LDL-C has been observed in this study, researchers are suggested to conduct additional clinical studies to explore the long-term benefits of these two approaches in reducing the risk of developing CVD among Chinese older adults.

## 5. Conclusions

Adopting a four-week MIND diet with four nutrition sessions on four consecutive weekdays or in addition to the MIND diet, practicing 2-h FB sessions on four consecutive weekends has been observed to have beneficial effects on cardiac and mental health in hypertensive Chinese older adults. The MIND diet and FB are new alternative approaches that are simple and affordable complementary interventions for hypertension management. This RCT study is the first to examine the synergistic effects of the MIND diet and FB on cardiac and mental health. The results may provide population risk evidence that adopting a MIND diet and FB is good for promoting cardiac and mental health well-being. The pilot study also confirmed the appropriateness of the intervention plan and outcome measurements for future controlled trials on this topic.

## Figures and Tables

**Figure 1 ijerph-19-14665-f001:**
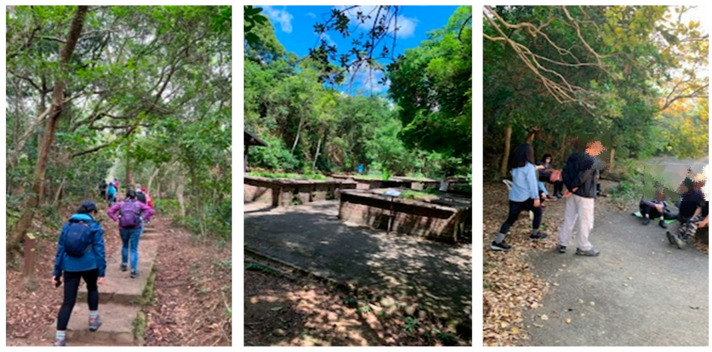
The scene of the designated path and the activities of the forest walk.

**Figure 2 ijerph-19-14665-f002:**
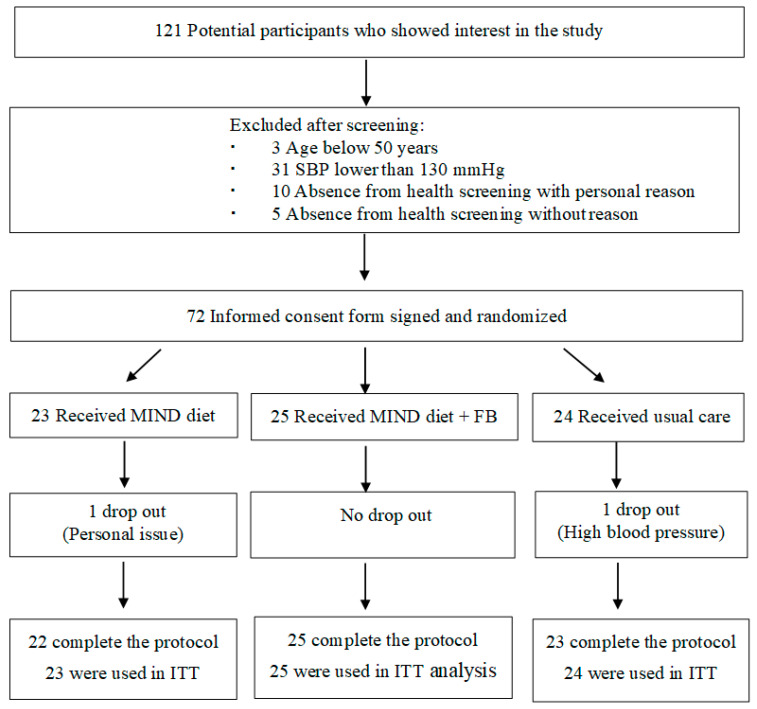
Flowchart of the participant selection process in the study period.

**Figure 3 ijerph-19-14665-f003:**
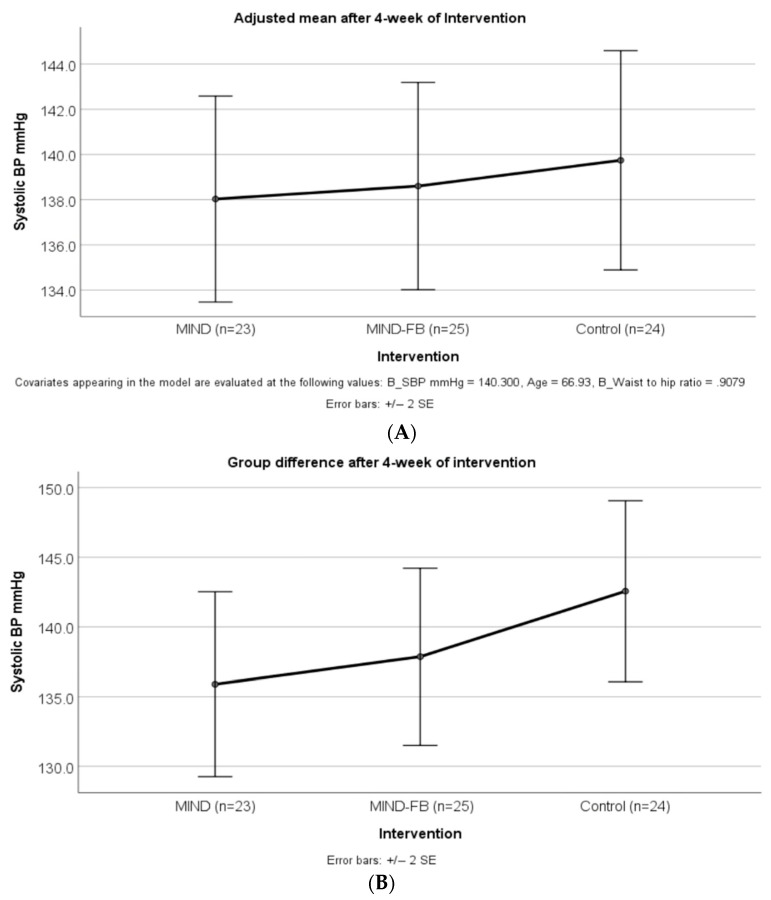
Comparison of means and 95% CI for SBP using an ANCOVA model after 4-week-intervention. The results were adjusted for age, waist to hip ratio, and pre-intervention BP. (**A**) showed the differences of SBP (*p* = 0.888) between MIND and MIND_FB groups and the control group were non-significant. (**B**) showed the difference of SBP (*p* = 0.339) between MIND and MIND_FB groups, and the control group was non-significant. SBP = systolic blood pressure; MIND group = The Mediterranean-DASH Intervention for Neurodegenerative Delay diet group; MIND-FB group = The Mediterranean-DASH Intervention for Neurodegenerative Delay (MIND) diet plus forest bathing group.

**Table 1 ijerph-19-14665-t001:** Schedule of each forest visit.

Time	Invitation Activities	Required Physical Activity
09:00–09:20	Share the forest experience	sitting or standing
09:20–09:30	Sharing circle	sitting or standing
09:30–09:50	Pleasant of moment by five senses	sitting or standing
09:50–10:00	Sharing circle	sitting or standing
10:00–10:20	Slow down physically and mentally	wandering to the designed forest path
10:20–10:30	Sharing circle	sitting or standing
10:30–10:50	Build up connection to forest	strolling around the designed path
10:50–11:00	Sharing circle	sitting or standing

**Table 2 ijerph-19-14665-t002:** Baseline socio-demographics and lifestyle characteristics of study subjects (ITT analysis).

		MIND Group (*n* = 23)	MIND-FB Group (*n* = 25)	Control Group (*n* = 24)	*p* Value across All Groups
Age Group					0.041
	<65 years	9 (39.1)	14 (56.0)	5 (20.8)	
	65 years or above	14 (60.9)	11 (44.0)	19 (79.2)	
Gender					0.636
	Female	19 (82.6)	19 (76.0)	17 (70.8)	
	Male	4 (17.4)	6 (24.0)	7 (29.2)	
Educational level					0.103
	Primary or below	8 (34.8)	3 (12.0)	12 (50.0)	
	Secondary	10 (43.5)	14 (56.0)	10 (41.7)	
	Associate degree	1 (4.3)	3 (12.0)	0 (0)	
	Bachelor’s degree	1 (4.3)	3 (12.0)	0 (0)	
	Master’s degree	3 (13.0)	2 (8.0)	1 (4.2)	
	Doctorate degree/PhD degree	0 (0)	0 (0)	1 (4.2)	
Current employment					0.676
	Employed Full time	3 (13.0)	5 (20.0)	5 (20.8)	
	Employed Part time	2 (8.7)	1 (4.0)	0 (0)	
	Homemaker	5 (21.7)	6 (24.0)	3 (12.5)	
	Retired	13 (56.5)	13 (52.0)	16 (66.7)	
Marital Status					0.559
	Married	15 (65.2)	15 (60.0)	12 (50.0)	
	Single/Divorced/Separated/Widow	8 (34.8)	10 (40.0)	12 (50.0)	
Frequency for exercise					0.266
	1 to 2 times a week	10 (43.5)	14 (56.0)	9 (37.5)	
	3 to 4 times a week	6 (26.1)	5 (20.0)	6 (25.0)	
	5 to 6 times a week	6 (26.1)	3 (12.0)	9 (37.5)	
	Never	1 (4.3)	3 (12.0)	0 (0)	
Times for exercise per session					0.184
	>40 min	10 (43.5)	5 (20.0)	8 (33.3)	
	30–40 min	3 (13.0)	9 (36.0)	3 (12.5)	
	20–30 min	3 (13.0)	3 (12.0)	8 (33.3)	
	10–20 min	3 (13.0)	2 (8.0)	3 (12.5)	
	<10 min	3 (13.0)	3 (12.0)	2 (8.3)	
	Never	1 (4.3)	3 (12.0)	0 (0)	
Smoking Habit					0.357
	Smoke 1–10 cigarettes a day	1 (4.3)	0 (0)	0 (0)	
	Ex-smoker	1 (4.3)	0 (0)	0 (0)	
	Never	21 (91.3)	25 (34.7)	24 (33.3)	
Drinking Habit					0.280
	More than fourteen times a week	1 (4.3)	0 (0)	0 (0)	
	Five to seven times a week	0 (0)	1 (4.0)	0 (0)	
	Once or twice a week	1 (4.3)	3 (12.0)	0 (0)	
	Ex-drinker	1 (4.3)	0 (0)	0 (0)	
	Never	20 (87.0)	21 (84.0)	24 (100)	
Relaxation practice					0.329
	Yoga	2 (8.7)	0 (0)	0 (0)	
	Tai chi or qigong	0 (0)	0 (0)	1 (4.2)	
	Stretching exercise	4 (17.4)	5 (20.0)	5 (20.8)	
	Spend time in nature	3 (13.0)	5 (20.0)	1 (4.2)	
	Slow deep breathing	3 (13.0)	7 (28.0)	4 (5.6)	
	Mindfulness meditation	1 (4.3)	1 (4.0)	2 (8.3)	
	Listen to music	6 (26.1)	5 (20.0)	3 (12.5)	
	Never	1 (4.3)	0 (0)	4 (16.7)	
	Others	3 (13.0)	2 (8.0)	4 (16.7)	
History of Hypertension					0.556
	Hypertension with medication	14 (60.9)	13 (52.0)	14 (58.3)	
	Hypertension without medication	0 (0)	1 (4.0)	1 (4.2)	
	Pre-hypertension	9 (39.1)	9 (36.0)	9 (37.5)	
	Unknown Hypertension	0 (0)	2 (8.0)	0 (0)	
No. of antihypertensive drugs taken					0.661
	0	9 (39.1)	13 (52.0)	10 (41.7)	
	1	8 (34.8)	9 (36.0)	8 (33.3)	
	2	4 (17.4)	1 (4.0)	5 (20.8)	
	3	1 (4.3)	2 (8.0)	1 (4.2)	
	4	1 (4.3)	0 (0)	0 (0)	

Data are presented as number (%). *p*-value was calculated by Pearson Chi-Square test; MIND group = The Mediterranean-DASH Intervention for Neurodegenerative Delay diet group; MIND-FB group = The Mediterranean-DASH Intervention for Neurodegenerative Delay (MIND) diet plus forest bathing group; ITT: intention to treat analysis.

**Table 3 ijerph-19-14665-t003:** Baseline measures of study subjects (ITT analysis for Continuous variables).

		MIND Group (*n* = 23)	MIND-FB Group (*n* = 25)	Control Group (*n* = 24)	*p* Value across All Groups	*p* Value *	
						MIND vs. Control	MIND-FB vs. Control
BMI (kg/m^2^)		24.9 ± 3.4	24.9 ± 3.3	25.7 ± 2.9	0.612	0.667	0.666
Waist to hip ratio		0.9 ± 0.04	0.89 ± 0.05	0.93 ± 0.05	0.014	0.080	0.015
Body fat (%)		35.1 ± 6.08	35.2 ± 6.9	37.3 ± 4.9	0.347	0.401	0.435
Lipid Panel							
	Total cholesterol (mmol/L)	5.4 ± 1.3	6.0 ± 1.4	5.5 ± 1.2	0.213	0.953	0.361
	LDL-cholesterol (mmol/L)	2.9 ± 1.2	3.4 ± 1.3	2.8 ± 1.1	0.249	0.979	0.276
	HDL-cholesterol (mmol/L)	1.7 ± 0.5	1.7 ± 0.3	1.8 ± 0.3	0.682	0.998	0.710
	Triglyceride (mmol/L)	1.8 ± 1.1	2.1 ± 1.1	2.0 ± 1.2	0.602	0.804	0.928
Glucose (mmol/L)		5.9 ± 1.3	6.2 ± 1.4	6.4 ± 1.6	0.652	0.640	0.960
BP							
	SBP (mmHg)	137.8 ± 14.3	141 ± 12.0	141.8 ± 17.2	0.609	0.611	0.977
	DBP (mmHg)	85.1 ± 10.5	88.5 ± 9.0	80.2 ± 9.7	0.014	0.199	0.011
Mood state							
	POMS total (score)	38.3 ± 13	41.8 ± 11.4	43.0 ± 14.8	0.456	0.444	0.938
	Tension-anxiety	4.1 ± 2.9	6.2 ± 3.9	5.5 ± 3.7	0.132	0.352	0.819
	Depression-dejection	3.2 ± 3.1	4.2 ± 3.5	4.7 ± 3.5	0.315	0.301	0.900
	Fatigue-inertia	6.3 ± 3.6	6.6 ± 3.7	6.8 ± 3.8	0.906	0.898	0.964
	Anger-hostility	4.0 ± 2.4	4.8 ± 2.6	4.1 ± 2.9	0.491	0.994	0.590
	Confusion-bewilderment	11.3 ± 3.0	10.6 ± 1.5	11.4 ± 3.1	0.542	0.988	0.562
	Vigor-activity	9.6 ± 2.8	9.5 ± 3.4	10.3 ± 4.6	0.709	0.785	0.726
Anxiety level							
	STAI-S (score)	40.4 ± 8.4	38.8 ± 8.5	36.5 ± 7.0	0.246	0.220	0.568
	STAI-T (score)	39.8 ± 9.0	42.9 ± 8.9	42.1 ± 9.5	0.499	0.679	0.951

Data are presented as mean ± SD. *p*-value were compared with one-way ANOVA. ANOVA = analysis of variance; BMI = body mass index; BP = blood pressure; MIND group = The Mediterranean-DASH Intervention for Neurodegenerative Delay diet group; MIND-FB group = The Mediterranean-DASH Intervention for Neurodegenerative Delay (MIND) diet plus forest bathing group; DBP = diastolic blood pressure; HDL = high-density lipoprotein; LDL = low-density lipoprotein; ITT = intention-to-treat analysis; SBP = systolic blood pressure; STAI = State and Trait Anxiety Inventory; POMS = The Profile of Mood States. * Tukey honest significant difference (HSD).

**Table 4 ijerph-19-14665-t004:** Changes in adiposity, blood pressure, cardiovascular risk factors, mood state, anxiety level, and MIND diet pattern (ITT analysis).

Variable	Mean Changes from Baseline to 4 Weeks After Intervention (95% CI)						*p* Value Across All Groups	Effect Size	MIND vs. Control		MIND-FB vs. Control	
	MIND Group (*n* = 23)	*p* Value #	MIND-FB Group (*n* = 25)	*p* Value #	Control Group (*n* = 24)	*p* Value #			Mean (95% CI) Between Group Difference	*p* Value *	Mean (95% CI) Between Group Difference	*p* Value *
Adiposity												
BMI (kg/m^2^)	−0.04 (−0.14 to 0.05)	0.347	−0.18 (0.39 to 0.01)	0.065	0.05 (−0.15 to 0.25)	0.604	0.013	0.239	−0.09 (−0.33 to −0.15)	0.44	−0.24 (−0.47 to −0.00)	0.047
Waist to hip ratio	−0.02 (−0.04 to 0.01)	0.156	0.00 (−0.01 to 0.01)	0.856	−0.01 (−0.03 to 0.00)	0.070	0.037	0.548	−0.00 (−0.02 to 0.02)	0.79	−0.01 (−0.01 to 0.03)	0.280
Body fat (%)	−0.08 (−0.70 to 0.54)	0.795	−0.69 (−1.50 to 0.12)	0.091	−0.48 (−2.36 to 1.39)	0.599	0.764	0.100	0.41 (−1.30 to 2.11)	0.64	−0.21 (−1.88 to 1.46)	0.800
Blood pressure												
SBP (mmHg)	−1.93 (−6.86 to 2.98)	0.422	−3.18 (−8.70 to 2.34)	0.246	0.67 (−3.11 to 4.44)	0.718	0.490	0.141	−2.61 (−9.24 to 4.03)	0.44	−3.85 (−10.3 to 2.65)	0.240
DBP (mmHg)	−1.26 (−4.51 to 1.99)	0.429	−2.57 (−4.94 to −0.18)	0.036	−0.27 (−2.44 to 1.90)	0.798	0.435	0.152	−0.99 (−4.59 to 0.61)	0.59	−2.29 (−5.82 to 1.24)	0.200
Cardiovascular risk factor												
Total cholesterol (mmol/L)	−0.61 (−0.85 to −0.37)	0.001	−0.86 (−1.34 to −0.37)	0.001	−0.58 (−0.92 to −0.42)	0.002	0.480	0.145	−0.03 (−0.55 to 0.49)	0.91	−0.28 (−0.79 to 0.23)	0.270
LDL-cholesterol (mmol/L)	−0.34 (−0.60 to 0.07)	0.015	−0.59 (−1.01 to −0.17)	0.008	−0.28 (−0.53 to −0.04)	0.026	0.329	0.179	−0.05 (−0.50 to 0.39)	0.81	−0.31 (−0.74 to 0.13)	0.160
HDL-cholesterol (mmol/L)	−0.24 (−0.38 to −0.10)	0.002	−0.14 (−0.24 to −0.05)	0.005	−0.19 (−0.29 to −0.09)	0.001	0.460	0.148	−0.05 (−0.21 to 011)	0.52	0.05 (−0.11 to 0.20)	0.550
Triglyceride (mmol/L)	−0.30 (−0.66 to 0.05)	0.089	−0.28 (−0.54 to −0.02)	0.036	−0.27 (−0.60 to 0.07)	0.116	0.985	0.000	−0.04 (−0.46 to 0.40)	0.86	−0.01 (−0.44 to 0.41)	0.950
Glucose (mmol/L)	−0.68 (−1.34 to −0.03)	0.042	−0.22 (−0.89 to 0.45)	0.507	−0.50 (−1.41 to 0.41)	0.269	0.667	0.110	−0.18 (−1.23 to 0.86)	0.73	0.28 (−0.74 to 1.30)	0.590
Mood states												
POMS total	−1.30 (−6.26 to 3.66)	0.591	−6.56 (−12.1 to 1.03)	0.022	0.13 (−4.20 to 4.45)	0.953	0.120	0.245	−1.43 (−8.03 to 5.44)	0.68	−6.69 (−13.4 to 0.04)	0.051
Tension–anxiety	0.04 (−1.14 to 1.23)	0.940	−2.20 (−3.98 to −0.42)	0.018	−0.38 (−1.61 to 0.86)	0.537	0.059	0.281	0.42 (−1.58 to 2.41)	0.68	−1.83 (−3.70 to 0.13)	0.070
Depression–dejection	0.17 (−0.80 to 1.25)	0.740	−1.36 (−2.96 to 0.24)	0.092	−0.13 (−1.47 to 1.22)	0.849	0.225	0.205	0.30 (−1.59 to 2.19)	0.75	−1.24 (−3.09 to 0.62)	0.190
Fatigue–inertia	−0.91 (−2.22 to 0.40)	0.162	−1.48 (−2.62 to 0.34)	0.013	−0.25 (−1.23 to 0.73)	0.604	0.292	0.187	−0.66 (−2.25 to 0.92)	0.41	−1.23 (−2.78 to 0.32)	0.120
Anger–hostility	−0.26 (−1.38 to 0.85)	0.633	−1.24 (−2.42 to −0.06)	0.039	1.08 (−0.36 to 2.53)	0.135	0.028	0.313	−1.34 (−3.08 to 0.39)	0.13	−2.32 (−4.20 to −0.63)	0.008
Confusion–bewilderment	−0.52 (−1.66 to 0.61)	0.351	−1.28 (−2.08 to −0.48)	0.003	−0.46 (−1.09 to 0.18)	0.149	0.305	0.184	−0.06 (−1.26 to 1.14)	0.92	−0.82 (−2.00 to 0.35)	0.174
Vigor–activity	−0.17 (−1.46 to 1.11)	0.781	0.64 (−0.66 to 1.94)	0.321	0.17 (−0.82 to 1.16)	0.732	0.611	0.118	−0.34 (−2.00 to 1.32)	0.68	−0.47 (−1.15 to 2.10)	0.560
Anxiety level												
STAI-S (score)	−1.78 (−4.29 to 0.73)	0.155	−5.28 (−8.14 to −2.42)	0.001	2.29 (−0.74 to 5.33)	0.132	0.001	0.431	−4.07 (−7.96 to −0.19)	0.04	−7.57 (−11.4 to −3.8)	0.000
STAI-T (score)	−0.78 (−4.31 to 2.74)	0.132	−4.28 (−8.06 to 0.50)	0.028	−0.75 (−2.79 to 1.29)	0.454	0.185	0.219	−0.03 (−4.47 to 4.41)	0.99	−3.53 (−7.88 to 0.82)	0.110

Remarks: # *p* < 0.05 for change during the study determined by paired *t*-test. * *p* value were compared with one-way repeated ANOVA using Tukey HSD; BMI = body mass index; MIND group = The Mediterranean-DASH Intervention for Neurodegenerative Delay diet group; MIND-FB group = The Mediterranean-DASH Intervention for Neurodegenerative Delay (MIND) diet plus forest bathing group; DBP = diastolic blood pressure; HDL = high-density lipoprotein; HSD = honest significant difference; LDL = low-density lipoprotein; ITT = intention-to-treat analysis; SBP = systolic blood pressure; STAI-S = state and trait anxiety inventory-state; STAI-T = state and trait anxiety inventory-trait; POMS = the profile of mood states; Effect size is calculated by r2 coefficient of determination.

## Data Availability

Not applicable.

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
