# Peer review of "Cardiac and Mental Benefits of Mediterranean-DASH Intervention for Neurodegenerative Delay (MIND) Diet plus Forest Bathing (FB) versus MIND Diet among Older Chinese Adults: A Randomized Controlled Pilot Study"

_ijerph, 2022, doi:10.3390/ijerph192214665_

Round 1

Reviewer 1 Report (Previous Reviewer 2)

The authors have greatly improved the quality of the publication.

This manuscript is a resubmission of an earlier submission. The following is a list of the peer review reports and author responses from that submission.

Round 1

Reviewer 1 Report

The impact of forest bathing on health could be derived through physiological and psychological changes. The authors appear to observe that forest bathing may produces additional psychological benefits on health which was not observed in the group receiving dietary intervention (MIND) – which is very interesting indeed. The results are nicely presented in tables, and the manuscript is well prepared and clearly written. Impact of forest bathing on health is still not well studied and intervention studies such as the one presented here can provide important evidence to inform the healthy ageing agenda in specific population settings (e.g. in Hong Kong).

Main comments:

The study aims to examine the combined effect of MIND diet and forest bathing on health, but I am not clear what the impact of forest bathing (if any) has on health on its own in this current setting. Why didn’t the authors examine the health impact of Forest bathing on its own by assigning a group of participants specifically to Forest bathing (without MIND diet intervention) in the study design? This could have significantly helped with the interpretation of the study results.

I am not convinced that results compared to the control group are easily interpretable, including the figure 3 on SBP – given the difference in confounding variables at baseline (eg. age). Do the authors have an explanation for why diet pattern and lipid levels have changed signficantly during the 4-week period in the Control group?

In general, it appears that the impact of both interventions on cardiometabolic measurements are less convincing compared to their impact on psychological indicators.

More details on the forest bathing session would be helpful. How many kilometres/ steps were covered, and were there hill climbing involved? Also, were participants having to follow a specific exercise timetable for walk, rest and relaxation during the 2-hour session? The authors should take care that the identity of participants were protected in any photos shown.

In order to better appreciate the impact of the dietary element of the intervention, more details on how diet scores were calculated are needed. Ie. Are the diet scores calculated through standardised food frequency questionnaire or food diary record?

Were the biochemistry measurements (Chol/ glucose/ TG) obtained controlled for fasting status (interval since last meal)?

Reviewer 2 Report

The manuscript with a long title, but useful for explaining the point of the study - Cardiac and mental benefits of Mediterranean-DASH intervention for neurodegenerative delay (MIND) diet plus forest bathing (FB) versus MIND diet among older Chinese adults: A randomized controlled pilot study- is clear, relevant for the field and presented in a well-structured manner.  The authors have clearly defined the strengths of this pilot study as well as the limitations.  The authors preliminary presented as a pilot study that forest bathing (FB) could enhance the Mediterranean-DASH intervention for neurodegenerative delay (MIND) diet on cardiac and mental benefits in the Chinese population. References are relevant, and experimental design is appropriate. Tables and figures properly show data. Tables are easy to understand and the data interpret appropriately and consistently throughout the manuscript.

I have one minor comment. An  introduction is too long, and some parts of the introduction could be move-in discussion,

Round 2

Reviewer 1 Report

There is no robust evidence from this study that demonstrates an effect of forest bathing or diet on cardiovascular health. Many of the conclusion statements remain confusing / misleading:

Line 380: “The major findings of this study were that the MIND diet alone and MIND-FB were associated with a lower risk of CVD after the completion of 4 weeks of intervention. Practicing FB can enhance the cardioprotective effect of the MIND diet, reducing anxiety levels and negative moods among older Chinese individuals with hypertension.” –this is not justified by results presented in the study. Health outcome (such as time to CVD events) was not measured as part of the study, and so have no direct evidence relating to CVD risk.

Line 19: “Total cholesterol, and low-density lipoprotein (LDL- C) were significantly decreased (p<0.01) in the three groups” What the authors really meant was that these markers were found decreased in all 3 groups, including in the control group.

Line 23: “The results provide preliminary evidence that FB could enhance the MIND diet on cardiac and mental benefits in the Chinese population.” – again this statement needs to be significantly revised.
